# MicroRNAs in Dystrophinopathy

**DOI:** 10.3390/ijms23147785

**Published:** 2022-07-14

**Authors:** Ahyoung Lee, Jiwon Moon, Jin Yu, Changwon Kho

**Affiliations:** 1Korean Medicine Institute, School of Korean Medicine, Pusan National University, Yangsan 50612, Korea; ahyoung.lee@pusan.ac.kr; 2Urban Agriculture Research Division, National Institute of Horticultural and Herbal Science, Jeollabuk-do 55365, Korea; moonjw85@korea.kr; 3Ginseng Division, Department of Herbal Crop Research, National Institute of Herbal Science, Chungbuk 27709, Korea; yujin8603@korea.kr; 4Division of Applied Medicine, School of Korean Medicine, Pusan National University, Yangsan 50612, Korea

**Keywords:** dystrophinopathy, Duchenne muscular dystrophy, microRNA, biomarker, therapeutic target

## Abstract

Duchenne muscular dystrophy (DMD) and Becker muscular dystrophy (BMD), which represent the range of dystrophinopathies, account for nearly 80% of muscle dystrophy. DMD and BMD result from the loss of a functional dystrophin protein, and the leading cause of death in these patients is cardiac remodeling and heart failure. The pathogenesis and progression of the more severe form of DMD have been extensively studied and are controlled by many determinants, including microRNAs (miRNAs). The regulatory role of miRNAs in muscle function and the differential miRNA expression in muscular dystrophy indicate the clinical significance of miRNAs. This review discusses the relevant microRNAs as potential biomarkers and therapeutic targets for DMD and DMD cardiomyopathy as examples of dystrophinopathies.

## 1. Introduction

Dystrophinopathies are a group of X-linked inheritance disorders characterized by loss of limbs, loss of respiratory and cardiac muscle strength, and destruction of nerve tissue. There are two main forms of dystrophinopathy: Duchenne muscular dystrophy (DMD), which develops in early childhood and presents with severe symptoms, and Becker muscular dystrophy (BMD), which develops late as a milder form [1]. These diseases are caused by the absence [2] or marked reduction [3] of the dystrophin protein, which results from a mutation in the DMD gene on the X-chromosome. Therefore, DMD and BMD occur more frequently in males than females. Females with a defective allele may also show very mild symptoms, but are at risk for a gradual increase in heart disease [4]. Globally, the prevalence of muscular dystrophy (MD) is estimated at 3.6 per 100,000 people [5]. The expected prevalence of DMD is 4.8 per 100,000 and 1.6 per 100,000 for BMD [5]. DMD/BMD accounts for more than 80% of all MD cases [6].

Heart abnormalities are common in many forms of MD (Table 1), in which dilated cardiomyopathy (DCM) is typical. DMD/BMD patients also present with an abnormal heart rhythm or arrhythmia, and sometimes localized hypertrophy. Approximately 10–20% of DMD patients die from heart failure or sudden death because of progressive heart disease, which is a significant cause of patient death, along with respiratory failure. Because of the successful management of respiratory complications and the steady increase in life expectancy of DMD patients, primary cardiac death in DMD patients is expected to increase. In particular, given that up to 50% of BMD patients die of heart failure, cardiac abnormalities may develop before skeletal problems [6]. In addition, females carrying either the DMD or BMD dystrophin mutation develop heart failure, even in the absence of any skeletal muscle defects [7].

Dystrophin is a long (110 nm) and slender cytoskeletal protein of approximately 400 kDa in size. It is predominantly expressed in skeletal and cardiac muscles [8]. Mutations in the dystrophin gene, including intragenic deletions (60–65% of cases), duplications (5–15% of cases), and point mutations, result in dystrophin dysfunction [9]. For DMD, mutations destroy the reading frame and produce a severely truncated dystrophin protein that breaks down rapidly. In contrast, in BMD, the reading frame for the dystrophin mutation is maintained and expressed, thus producing a partially functioning protein [9]. Dystrophin is located on the inner surface of the sarcolemma. The N-terminal domain of dystrophin directly binds to the F-actin cytoskeleton [10], whereas the C-terminal cysteine-rich region of dystrophin interacts with a component of the sarcolemmal (glyco) protein complex [11], which in turn connects to the extracellular matrix [12]. This dystrophin-associated protein complex (DPC) participates in force transduction and stabilizes the plasma membrane of muscle cells during contraction. Therefore, either the absence of the dystrophin protein (DMD) or the reduced expression of dystrophin (BMD) causes the sarcolemma to become fragile with reduced stiffness and increased leakiness, rendering muscle cells susceptible to damage during contraction−relaxation [13]. In addition, growing evidence suggests DPC acts as a scaffold for signaling pathways in both skeletal and cardiac muscles [14,15]. Thus, DPC abnormalities are considered an important pathologic defect that causes muscle degeneration and cardiomyopathy.

Essential biological processes, including cell proliferation, differentiation, and apoptosis, are regulated, in part, by microRNAs (miRNAs) through post-transcriptional regulation of the gene expression [16]. Thus, miRNA dysregulation has been implicated in a variety of diseases. This review provides an overview and update on relevant miRNAs associated with dystrophinopathy, with a focus on DMD. Understanding the role and impact of miRNAs on DMD pathogenesis will provide insight into their potential utility (as tools and targets) for novel dystrophinopathy treatments.

## 2. microRNAs Associated with Dystrophinopathy

### 2.1. Importance of miRNA

MiRNAs are small, non-coding RNAs of approximately 22 nucleotides in length [17]. miRNA genes can be included in other genes or can be located in the intron of host genes. Sometimes, miRNA genes are composed of polycistronic clusters and are expected to be co-expressed [18]. The production of mammalian miRNAs is a highly regulated process [17]. In the nucleus, miRNA is transcribed by RNA polymerase II/III to generate primary miRNA (pri-miRNA), which are modified with a 7-methylguanosine cap structure and polyadenylation. Pri-miRNA is usually more than one kilobase long and contains a double-strand area incompletely in the hairpin loop. Through the canonical Drosha/DGCR8 cleavage or non-canocical pathways, pri-miRNA is converted to a hairpin-like precursor miRNA (pre-miRNA) with a length of about 70–100 nt. This pre-miRNA is exported from the nucleus to the cytoplasm with the help of Exportin 5, and is further processed by the RNase called Dicer to generate mature miRNA of ~22 nt in length. Each pre-miRNA can release two mature miRNA strands (5p and 3p) containing different messenger RNA (mRNA)-targeting sequences. In general, only one single strand becomes a template loaded into the RISC (RNA-induced silencing complex) to control the fate of specific mRNAs. The mature miRNA functions by complementary base-pairing with the miRNA response elements (MRE) located in the target mRNA. The 3′-untranslational region (3′-UTR) often contains MREs [19]. As a result, it can interfere with the translation of the transcript or can cause the degradation of the mRNA target directly (Figure 1).

In vertebrates, miRNAs function to modify or maintain cellular phenotypes during various types of stress, usually by repressing or regulating the expression of proteins that are specific to a particular cell type [20,21]. The human genome encodes approximately 2300 mature miRNAs [22]. Moreover, bioinformatic analyses suggest that a single miRNA can target up to 200 different genes and that a combination of miRNAs can regulate the expression of one-third of all human genes [23]. In addition, several studies have demonstrated an association of dysregulated miRNA expression with the pathological basis of various human diseases. Therefore, it is not surprising that miRNAs have important implications for human health and cellular phenotypes, and modulating miRNAs represents a novel therapeutic strategy, especially for multifactorial diseases for which there are currently no effective therapies.

An important characteristic of miRNAs is their presence in biofluids, which often correlates with various physiological states. Moreover, the robust chemical stability and development of sensitive detection methods (e.g., next-generation sequencing, real-time PCR, and microarrays) support the potential use of miRNAs as biomarkers. After the first tumor-associated miRNAs (e.g., miR-21, miR-155, and miR-210) were discovered in the serum of cancer patients [24], the list of miRNAs as biomarkers has grown significantly. Elevated serum levels of a muscle-specific isoform of creatine kinase (CK) in DMD, along with genetic assessments, are widely used diagnostic markers. However, serum CK is neither DMD-specific nor a strong predictor of cardiac function [25]. Therefore, circulating miRNAs have been proposed as biomarkers for the diagnosis and prognosis of DMD [26].

### 2.2. miRNAs as a DMD Biomarker

MiRNAs are expressed with tissue specificity. Eight miRNAs, namely miR-1, miR-133a, miR-133b, miR-206, miR-208a, miR-208b, miR-499a, and miR-499b, were identified as muscle-specific miRNAs [27,28,29,30]. Except for miR-206 (predominant in skeletal muscle) and miR-208a (predominant in cardiac muscle), they are expressed in both cardiac and skeletal muscle tissues. Three miR families, miR-1, miR-133, and miR-206, are among the most abundant in muscle cells, accounting for more than 25% of all miRNAs [29]. They are involved in skeletal muscle proliferation and differentiation [31,32].

Several microarray analyses have identified DMD-associated miRNAs [33,34,35]. For example, muscle-specific miRNAs were found to be differentially expressed in dystrophic muscle tissues in patients and in mdx mice, a well-established mouse model for DMD research. The miRNAs, miR-1, miR-133, and miR-206, were decreased in the dystrophic muscles compared with healthy tissue. Moreover, a significant upregulation (~70-fold) of miR-31 inhibited the dystrophin expression by targeting the 3′-UTR of dystrophin in DMD muscles [36]. Importantly, several muscle-enriched miRNAs (miR-1, miR-133, miR-206, miR-208, and miR-499) were upregulated in the dystrophic sera, not only in animal models, but also in patients (Table 2). In particular, the expression of miR-1 and miR-133, as a result of muscle degeneration, was highly elevated in the dystrophic serum of DMD patients (up to 100-fold in DMD and up to 30-fold in BMD versus healthy controls) [37]. Furthermore, upregulated serum miR-206 in female carriers of DMD, who were in most cases asymptomatic, were reported, suggesting a potential use in carrier detection [38]. In addition, a close relationship was reported between the levels of many serum miRNAs and the functional performance of DMD patients [39]. 

Several non-muscle-specific miRNAs, including miR-30c, miR-181a, and miR-95, were also elevated in the serum or plasma of DMD patients [40,41]. Compared with healthy individuals over 4 years, the serum levels of miR-30c and miR-206 were significantly elevated in BMD/DMD patients [42]. Moreover, the miR-206 levels clearly exhibited characteristics that were different between patients with BMD and DMD. Recent clinical interventions for DMD, such as antisense-mediated exon skipping, suggest an urgent need for reliable biomarkers. Interestingly, the serum levels of miR-1, miR-133, and miR-206 in the mdx mice were correlated with the exon-skipping activity-dependent dystrophin expression [35]. However, the expressions of these miRNAs were not significantly different in the DMD patient serum before and after exon-skipping (Eteplirsen) treatment in a small number of samples [39].

**Table 2 ijms-23-07785-t002:** Muscle-enriched miRNAs elevated in dystrophic serum.

	Analysis Samples	Function	Refs.
miR-1	DMD patients, DMD mice, DMD dogs	Myogenesis	[37,43]
miR-133	DMD patient, DMD mice, DMD dogs	Myogenesis	[37,43]
miR-206	DMD patient, DMD female carrier, DMD mice, DMD dogs	Muscle development & regeneration	[37,38,43]
miR-208b	DMD patient, DMD mice, DMD dogs	Muscle fiber determination, Myogenesis	[41,44]
miR-499	DMD patient, DMD mice	Muscle fiber determination	[44]

### 2.3. microRNAs Associated with DMD Cardiomyopathy

MiRNAs are involved in maintaining the cardiac structure and function. In fact, knock out mice for Dicer, an enzyme that produces short RNA fragments, or Dgcr8, an enzyme involved in the early stages of miRNA biosynthesis, exhibit dilated cardiomyopathy, heart failure, and premature death [45,46]. In addition to expression profiling, animal studies have shown that several muscle-specific DMD-related miRNAs, such as miR-1, miR-133, and miR-208, are involved in cardiac muscle remodeling and pathogenesis (Table 3).

miR-1: Several studies have demonstrated a pathophysiological role for miR-1 in heart disease. For example, an abnormal expression of miR-1 is involved in electrical remodeling, such as the development of arrhythmias. With respect to the underlying mechanism, miR-1 directly targets the key proteins that regulate potassium current (e.g., potassium ion channel and gap junction protein) [47] or proteins involved in muscle cell Ca^2+^ cycling (e.g., protein phosphatase PP2A) [48]. In addition, miR-1 is associated with mechanical remodeling, suggesting that it has anti-hypertrophic properties. A reduced miR-1 expression along with miR-133 has been reported in three different mouse models of cardiac hypertrophy [49]. Because miR-1 is transcribed as a bicistronic transcript, together with members of the miR-133 family, it raises the possibility of cooperation in cardiac hypertrophy. In addition, miR-1 overexpression was shown to attenuate agonist-induced cardiac hypertrophy in vitro and in vivo [50].

miR-133: MiR-133 has been implicated in myocardial remodeling. Similar to miR-1, miR-133 overexpression suppresses cardiac cell hypertrophy [49]. In contrast, miR-133 inhibition induces cardiac hypertrophy by targeting cytoskeletal and myofibrillar rearrangement-related proteins (e.g., RhoA and CDC42) [49]. Decreased miR-133 promotes the progression of cardiac fibrosis. Several central mediators in tissue fibrosis, such as TGF-β1 (transforming growth factor-β1), CTGF (connective tissue growth factor), and COL1A1 (collagen type 1-alpha 1), have been identified as direct targets of miR-133 [51,52,53]. The accumulation of these miR-133 target molecules contributes to collagen deposition and fibrosis. Interestingly, miR-133 can be directly regulated by other noncoding RNA, such as linc-MD1. A muscle-specific long non-coding RNA (lncRNA), linc-MD1, binds to miR-133a and acts as a competitor RNA for targets of miR-133, including MAML1 (Mastermind-like 1) and MEF2C (Myocyte Enhancer Factor 2C) transcription factors [54]. The expression of linc-MD1 is diminished in Duchenne patient myoblasts. Although the role of lncRNAs in DMD pathogenesis is still unclear, understanding the lncRNAs−miRNAs−mRNAs network is important to explore the molecular mechanism of DMD.

miR-208a: The miR-208 family, which contains miR-208a/b and miR-499, contributes significantly to cardiac hypertrophy and arrhythmias. These three miRNAs are located in the introns of the genes encoding myosin heavy chain isoforms, which regulate the expression of sarcomeric contractile proteins [30,55]. Transgenic mouse studies have shown that miR-208a, a heart-enriched miRNA, is associated with hypertrophic cardiomyocyte growth and the upregulation of hypertrophy-related genes by targeting THRAP1 (thyroid hormone receptor-associated protein 1) and myostatin 2 [55]. Cardiac conduction abnormalities have also been reported in both miR-208a deletion mice and miR-208 overexpressing mice [28,55]. In addition, miR-208a null mice exhibited a reduction in cardiac contractility [36]. As miR-208a can be released from heart muscle cells into the serum and plasma in response to cardiac pathogenesis, several studies have evaluated their diagnostic value. For example, the sensitivity and specificity of circulating miR-208a were reported in patients with myocardial damage [56], severe COVID-19 [57], and heart failure with reduced ejection fraction [58].

miR-339-5p: It was recently shown that miR-339-5p is upregulated and released by exosomes from DMD patient-induced pluripotent stem cell-derived cardiomyocytes (DMD-iCMs). Downregulation of miR-339-5p directly modulated stress-response genes and reduced cardiomyocyte death in DMD-iCMs. These data indicate a pathological role of elevated miR-339-5p in DMD cardiomyocytes [59].

**Table 3 ijms-23-07785-t003:** Cardiopathological relevance of muscle-specific DMD-related miRNAs.

	Expression	Validated Targets	Function	Refs.
miR-1	Down in HT/HF	PP2A, KCNJ2, GJA1, MEF2, HAND2	Ca^2+^ homeostasis Cardiac hypertrophy	[49]
miR-133	Down in HT/HF	RhoA, CDC42, CTGF, TGF-β1, COL1A1	Cardiac hypertrophy Myocardial fibrosis	[49,51,52,53]
miR-208a	Up in DCM	THRAP1, MSTN	Myocardial fibrosis Cardiac hypertrophy	[28]

CDC42, Cell division control protein 42 homolog; COL1A1, collagen type 1-alpha 1; CTGF, connective tissue growth factor; DCM, dilated cardiomyopathy; GJA1 gap junction protein alpha 1; HAND2, Heart- and neural crest derivatives-expressed protein 2; HT, hypertrophy; HF, heart failure; KCNJ2, potassium inwardly rectifying channel subfamily J; MEF2, myocyte enhancer factor-2; MSTN, Myostatin; RhoA, Ras homolog gene family member A; THRAP1, thyroid hormone receptor-associated protein 1; TGF-β1, transforming growth factor-β1; PP2A, protein phosphatase 2A.

Serum miRNAs in disease carriers: Even without skeletal muscle symptoms, heart symptoms often occur in female DMD/BMD carriers for the asymptomatic form with mild abnormalities to progressive heart failure [60] and dilated cardiomyopathy [61], which may require heart transplantation [62]. Therefore, early detection of heart disease in female carriers is important. Changes in the miRNA levels associated with heart and/or skeletal muscle pathologies, including cardiac hypertrophy (e.g., miR-22 and miR-26a), fibrosis (e.g., miR-26a, miR-222, and miR-378a-5p), muscle cell death (e.g., miR-342), and regulation of skeletal muscle mass (e.g., miR-378 and miR-29c) regulators have been detected in the biofluids of disease carriers (Table 4) [63,64,65]. Interestingly, a significant downregulation of miR-29c was only found in the blood of female DMD carriers with cardiac symptoms detected by cardiovascular magnetic resonance [63]. It is worth determining whether this miR-29c downregulation is female-specific and whether its expression is comparable to healthy controls before cardiomyopathy in men with DMD.

## 3. microRNAs with Therapeutic Potential

### 3.1. Restoration of Dystrophin or Utrophin Expression

Recently, several therapies that restore dystrophin expression have been successfully developed for clinical studies. In addition, miRNAs that mediate an increase of dystrophin or utrophin, an autosomal paralogue of dystrophin, have attracted attention as therapeutic candidates.

miR-31: MiR-31 is expressed in regenerating fibers, which are activated at the onset of DMD in both mouse and human muscles [36]. DMD myoblasts with accumulated miR-31 exhibit a lower differentiation potential and miR-31 has been shown to target dystrophin mRNA [36]. Importantly, the inhibition of miR-31 function may improve the therapeutic efficacy of restoring the dystrophin expression. In human DMD myoblasts, miR-31 inhibition increased dystrophin synthesis with exon 51 skipping [36]. In the skeletal muscle, miR-31 also regulates myogenesis by inhibiting MYF5 (myogenic factor 5), an activator of muscular satellite cells, which are identical to myogenic stem cells [66]. Thus, miR-31 inhibition represents a therapeutic strategy to improve DMD muscle function by improving dystrophin synthesis and muscle differentiation. Furthermore, miR-31 has a role in cardiac remodeling. For example, atrial miR-31 inhibition contributes to electrical remodeling, such as the termination of atrial fibrillation, by the restoring dystrophin and nNOS (neuronal nitric oxide synthetase) levels [67]. The cardioprotective effects of miR-31 silencing have also been reported in rats with myocardial infarction [68]. Mechanistically, miR-31 represses TNNT2 (troponin T2), E2F6 (E2F Transcription Factor 6), NR3C2 (Nuclear Receptor Subfamily 3 Group C Member 2), and TIMP4 (Metalloproteinase inhibitor 4) by binding to their respective target 3′-UTR sequences.

miR-206: Skeletal muscle-specific miR-206 is considered a new target for DMD therapy because of its ability to regulate the expression of utrophin. In humans, utrophin is nearly 80% identical to dystrophin [69] and it is naturally increased in the sarcolemma of dystrophic skeletal muscles as a compensatory mechanism for dystrophin deficiency in mice [70,71] and humans [72]. In mdx mice, downregulation of miR-206 resulted in a higher utrophin expression and an improvement in the dystrophic phenotype [73]. In addition to miR-206, the suppression of utrophin by several miRNAs, including let-7c, miR-150, miR-196b, miR-296-5p, and miR-133b, was demonstrated [74,75]. A therapeutic strategy to enhance utrophin production could be applicable to BMD patients, as well as all DMDs, regardless of the type of dystrophin mutation. Currently, several drugs that stabilize the utrophin−glycoprotein complex, including TVN-102 (recombinant human biglycan) and rhAKM111 (recombinant human protein laminin-111), have been developed for utrophin therapy [76]. Meanwhile, miR-206 has been characterized as a cardioprotective molecule. It is upregulated in response to stress and promotes the survival of heart muscle cells in vitro and in vivo [77].

### 3.2. Inhibition of Pathogenic Muscle Remodeling: Anti-Fibrosis

Muscle fibrosis, which is a massive accumulation of connective tissues, occurs as a result of chronic tissue damage and inflammation, and is associated with muscle wasting in DMD/BMD [78]. In addition to suppressing muscle regeneration, fibrosis accelerates disease progression by disturbing proper therapy and by altering metabolism. There are two well-known miRNAs that are associated with fibrosis in DMD conditions.

miR-21: MiR-21 is upregulated in DMD fibroblasts, the major collagen-producing cells, and is correlated with the expression of pro-fibrotic genes [79]. Histological evaluations have shown that the inhibition of miR-21 reduces diaphragmatic fibrosis in mdx mice. In addition, in the diaphragm where miR-21 was overexpressed, the soluble collagen content was reduced compared with the control group [79]. MiR-21 is also tightly linked to cardiac fibrosis. Elevated miR-21 promotes cell transition from cardiac fibroblasts to myofibroblasts, resulting in pathogenic heart modeling [80]. MiR-21 is considered a potential therapeutic target for the treatment of cardiomyopathies [81]. With respect to fibrosis, Regulus therapeutics is currently conducting a phase 2 clinical trial jointly with Sanofi Genzyme to evaluate the safety and efficacy of anti-miR-21 oligonucleotides (RG-012) in Alport patients (ClinicalTrials.gov Identifier: NCT02855268).

miR-29: MiR-29 has pro-myogenic activity and anti-fibrotic properties in DMD. Downregulation of all three members of the miR-29 family, miR-29a, miR-29b, and miR-29c, were detected in dystrophic mouse muscles, including the limb muscles, diaphragm, and heart [82]. A reduced miR-29a and miR-29c expression has also been found in human DMD myoblasts [79]. Systemic delivery of miR-29 mimics into mdx mice promotes diaphragm muscle regeneration and inhibits the development of fibrosis in the diaphragm by directly repressing extracellular matrix components, such as microfibrillar-associated protein 5 and collagen [82]. The overexpression of miR-29c reduces muscle fibrosis in dystrophin and utrophin double-knockout mice [83]. The co-overexpression of miR-29 in DMD mice with micro-dystrophin restores fibrosis and muscle function to a similar levels to that of normal mice, suggesting a possible combination treatment strategy [83]. The therapeutic effects of miR-29 have been extensively studied in multiple organ fibrosis, including the heart [84]. Currently, the oligonucleotide mimic of miR-29b (Remlasten or MRG-201) is in phase 2 clinical trials for cutaneous fibrosis (ClinicalTrials.gov Identifier: NCT03601052).

### 3.3. Restoration of Abnormal Calcium Homeostasis

Excessive intracellular Ca^2+^ levels are a major secondary pathologic event in dystrophic muscles. For DMD, stretch-induced stress results in membrane tears that induce extracellular Ca^2+^ influx and cellular Ca^2+^ accumulation. Furthermore, abnormal Ca^2+^ cycling between the cytosol and sarcoplasmic reticulum (SR)/endoplasmic reticulum (ER) contributes to Ca^2+^ overload in dystrophic muscles [85]. This disturbance in Ca^2+^ homeostasis eventually contributes to DMD pathology [85]. Several mechanisms have been proposed for Ca^2+^-mediated muscle disease and DMD cardiomyopathy. Aberrant Ca^2+^-dependent protein degradation results in proteolytic damage to cellular proteins and myofibrillar proteins [17]. The activity of Ca^2+^-activated proteases, including calpain, is elevated in the muscles of DMD patients [86,87]. The activation of calpain contributes to the breakdown of myofibrillar proteins and eventually impairs muscle function [88,89]. Cellular Ca^2+^ overload is also associated with necrosis and apoptotic pathways in DMD [90,91]. Abnormalities in ER/SR Ca^2+^ handling proteins, including reduced sarco/endoplasmic reticulum Ca^2+^-ATPase (SERCA) pump activity and the hypersensitive ryanodine receptor (RyR) channel, gradually increase the intracellular Ca^2+^ load and inhibit the contractile function of muscle cells [92,93]. This leads to repeated cycles of muscle degeneration/regeneration, loss of myocytes, inflammatory responses, fibrosis, and, consequently, progressive muscle weakness and dysfunction (Figure 2). Several preclinical animal studies have indicated that the normalization of Ca^2+^ abnormalities has therapeutic effects on DMD. In particular, targeting SR Ca^2+^ handling proteins, such as SERCA and RyR, shows promise, and there are miRNAs that can directly regulate their expression.

miR-1: RyR stabilization may prevent Ca^2+^ leakage from the SR, thereby reducing intracellular Ca^2+^ accumulation. For example, pharmacological stabilization of the RyR Ca^2+^ release channel attenuates the disease phenotype in mdx mice [94]. In addition, treatment with RyR stabilizing molecules increased the efficacy of exon-skipping drugs in DMD cell culture models [95]. With respect to microRNAs, elevated miR-1 can hyperactivate the RyR2 channel by inhibiting the PP2A regulatory subunit B56α, which is a scaffold for the RyR2 complex [48].

miR-25: Overall, the SERCA pump overexpression has shown beneficial effects in dystrophic mice. Importantly, severe dilated cardiomyopathy was ameliorated by SERCA2a gene transfer in aged mdx mice [96]. Several miRNAs, including miR-25, have been identified that target SERCA2a mRNA. The therapeutic potential of inhibiting miR-25 has also been evaluated in a mouse model of heart failure [97]; however, the expression profile and muscle-related function of miR-25 in DMD have not been reported. Clinical studies of SERCA2a gene therapy have been conducted for heart failure [98,99], which may serve as an important basis for designing DMD and DMD cardiomyopathy therapies. Moreover, given the therapeutic implications of targeting Ca^2+^ cycling, in-depth studies of specific miRNAs that regulate Ca^2+^ mishandling in DMD are warranted. In particular, research on miRNAs that regulate calcium metabolism in heart disease is being actively conducted [100].

## 4. Conclusions

MiRNA has emerged as a key molecule involved in muscle gene expression and has expanded its utility as a biomarker in dystrophinopathy. Several muscle-specific miRNAs have been extensively examined for their potential as non-invasive biomarkers for DMD diagnosis and/or disease monitoring. In addition, miRNAs were identified that were associated with cardiomyopathy, the leading cause of death in DMD. The development of miRNA-based biomarkers will be of clinical significance, as many new therapeutic regimens for DMD have been tested in recent years. Meanwhile, the approval of Onpatro (Patisiran), the first small interfering RNA (siRNA)-based drug for the treatment of nerve damage caused by hereditary transthyretin-mediated amyloidosis, raises the prospect of miRNA as a therapeutic agent for genetic diseases. In DMD, miRNAs may be useful for several therapeutic strategies, such as for the stimulation of dystrophin or utrophin synthesis, the reduction of fibrosis, and the improvement of cardiac abnormalities. However, research has focused on only a few muscle-enriched miRNAs, and there are many more miRNAs involved in muscle remodeling, fibrosis, and Ca^2+^ cycling that have not yet been studied in dystrophinopathies. A further understanding of the dystrophinopathy-associated miRNA function will improve miRNA therapeutics in muscular dystrophies.

## Figures and Tables

**Figure 1 ijms-23-07785-f001:**
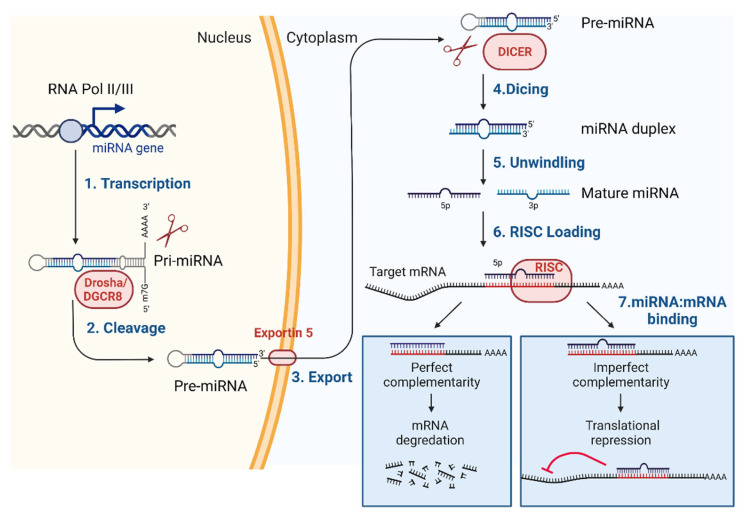
Overview of canonical miRNA biogenesis and function. MiRNA genes are expressed as long transcripts (called pri-miRNAs) that are transcribed by RNA polymerase II or III, and produce a precursor form of miRNA (called pre-miRNA) in the nucleus. The pre-miRNA is then exported to the cytoplasm by exportin 5 and is further processed by the Dicer complex to generate mature miRNA duplexes. Finally, mature miRNA duplexes are separated, and one of the strands is loaded into RISC (RNA silencing complex), which binds to the target mRNA. Mature miRNA regulates the specific gene expression by directly controlling the stability of the mRNA targets or suppressing translation. DGCR8; DiGeorge Syndrome Critical Region 8, m7G; 7-methylguanosine cap structure. Adapted from “microRNA in cancer” by BioRender.com (accessed on 29 June 2022). Retrieved from https://app.biorender.com/biorender-templates (accessed on 29 June 2022). Agreement number is MK245ND69F.

**Figure 2 ijms-23-07785-f002:**
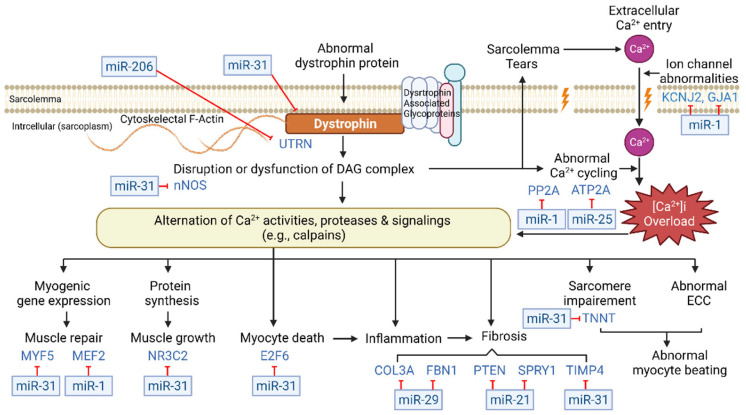
Role of calcium in the pathogenesis of dystrophinopathy. The figure includes the above-mentioned therapeutic candidate microRNAs and their target genes. An abnormal dystrophin expression may contribute to a loss of cytoskeletal and sarcolemma integrity and impaired calcium homeostasis. Cytoplasmic calcium overload plays a pivotal role in the disease progression of dystrophic skeletal and cardiac muscles. BMD; Becker muscular dystrophy, DAG; dystrophin-associated glycoproteins; DMD; Duchenne muscular dystrophy, ECC, excitation-contraction coupling.

**Table 1 ijms-23-07785-t001:** Examples of different types of muscular diseases associated with cardiac abnormalities. OMIM accessed on 1 May 2022.

Name	OMIM No.	Gene	Inheritance	Age of Onset (Years)	Cardiac Complications
Becker	300376	DMD	XLR	5 to 60	Arrhymias, DCM, HT
Duchenne	310200	DMD	XLR	2 and 3	Arrhymias, DCM, HT
Emery-Dreifuss	310300 (TP1) 181350 (TP2) 616516 (TP3)	EMD LMNA LMNA	XLR AD AR	10	AF, Arrhymias, AV block, DCM, Sudden death
Limb-Girdle	603511 (TP1) 253600 (TP2)	DNAJB6 CSPN3	AD AR	10 to 30	DCM in certain sub-type, Conduction disorders
Myotonic	160900 (TP1) 602668 (TP2)	DMPK CNBP	AD AD	20 to 40	AF, Arrhymias, AV block, DCM, Sudden death

AD, autosomal dominant; AF, atrial fibrillation; AV, atrioventricular; CSPN3, calpain-3; CNBP, CCHC-type zinc finger nucleic acid binding protein; DCM, dilated cardiomyopathy; DNAJB6, DnaJ heat shock protein family (Hsp40) member B6; DMD, dystrophin; DMPK, myotonic dystrophy protein kinase; EMD, emerin; HT, hypertrophy; LMNA, lamin A/C; OMIM, online mendelian inheritance in man; TP, Type; XLR, X-linked recessive.

**Table 4 ijms-23-07785-t004:** Circulating miRNAs in DMD/BMD-associated cardiomyopathy.

	Name	Source	Function	Refs.
Up	miR-22	DMDc plasma	Hypertrophy	[63]
	miR-26a	DMDc plasma DMDms/BMDms plasma	Fibrosis, Hypertrophy	[63,64]
	miR-206	DMDc plasma	Muscle regeneration	[63]
	miR-222	DMDms/BMDms plasma	Fibrosis	[64]
	miR-342	DMDc/BMDc plasma	Cardiomyocyte apoptosis	[63]
	miR-378a-5p	DMDc plasma DMDms/BMDms plasma	Fibrosis	[63,64]
	miR-378a-3p	DMDc/BMDc plasma	SKM mass regulator	[63]
Down	miR-29c	DMDc/BMDc plasma DMD urine	SKM mass regulator	[64,65]

DMDc/BMDc, DMD/DMD female carriers; DMD/BMDms, DMD/BMD with myocardial scars; SKM, skeletal muscle.

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
