# Peer review of "MicroRNAs in Dystrophinopathy"

_ijms, 2022, doi:10.3390/ijms23147785_

Round 1

Reviewer 1 Report

The manuscript summarizes the new findings on miRNAs in dystrophinopahty disease. The review is well-written and provides a comprehensive picture. Below my concerns

- In the title “the heart” is unnecessary since Dystrophinopathy already includes heart phenotype and no other heart-related disease are presented.

- Calcium role in DMD is well explained in the introduction but is perhaps too large with the risk of being off topic. For this reason, the 3.3 part on calcium homeostasis should be more concise and more miRNA-focused instead of about other molecular players.

- 2.1 part should include a brief description of miRNA biogenesis and mechanism of action. miRNAs are not “single-stranded hairpin RNA” but rather small ss RNAs. Primary and precursor miRNAs have the hairpin structures.

- miR 339 5p in part 2.3 could have a small bold title like mir1, 133, and 208a. The same for the part on disease female carriers.

- Part 4 is off topic and not correlated to the main body of the review neither for muscle and dystrophies nor for related molecular pathways. This part should be excluded.

Author Response

RESPONSE TO THE COMMENTS OF REVIEWERS

The manuscript summarizes the new findings on miRNAs in dystrophinopahty disease. The review is well-written and provides a comprehensive picture. Below my concerns

- In the title “the heart” is unnecessary since Dystrophinopathy already includes heart phenotype and no other heart-related disease are presented.

Ans) The title was revised according to the opinion of the reviewer.

- Calcium role in DMD is well explained in the introduction but is perhaps too large with the risk of being off topic. For this reason, the 3.3 part on calcium homeostasis should be more concise and more miRNA-focused instead of about other molecular players.

Ans) In this revision, the description of calcium abnormalities in DMD was transferred from the introduction to Part 3.3. It also significantly reduced the amount of descriptions associated with calcium modulating molecules. A new figure 2 was also presented, combining the original figures 1 and 2.

- 2.1 part should include a brief description of miRNA biogenesis and mechanism of action. miRNAs are not “single-stranded hairpin RNA” but rather small ss RNAs. Primary and precursor miRNAs have hairpin structures.

Ans) A brief description of miRNA biogenesis and mechanism of action was added to Part 2.1. As a result, two new citations were added. In addition, a new figure describing miRNA biogenesis and function was presented in Figure 1.

- miR 339 5p in part 2.3 could have a small bold title like mir1, 133, and 208a. The same for the part on disease female carriers.

Ans) The subtitle was added according to the reviewer’s suggestion.

- Part 4 is off topic and not correlated to the main body of the review neither for muscle and dystrophies nor for related molecular pathways. This part should be excluded.

Ans) Part 4 was deleted at the reviewer's request.

Reviewer 2 Report

This manuscript summarized for miRNAs as biomarkers in cardiac failure of DMD. This review showed very important  miRNAs for DMD clearly. But some corrections may be needed. As for the target genes for miRNAs in DMD, it is better to show the molecular mechanisms of DMD using figure.  In addition, it was better to add forms of these miRNAs in blood of DMD such as exosome-capsulated or RNA-binding miRNAs. In addition of the miRNAs,  it is better to add long non-coding RNAs corresponding to these miRNAs in DMD. 

Author Response

RESPONSE TO THE COMMENTS OF REVIEWERS

This manuscript summarized for miRNAs as biomarkers in cardiac failure of DMD. This review showed very important miRNAs for DMD clearly. But some corrections may be needed.

- As for the target genes for miRNAs in DMD, it is better to show the molecular mechanisms of DMD using figure. 

Ans) In the revised manuscript, a new figure 2 was provided, which is derived from the original figures 1 and 2. At the reviewer's request, a new Figure 2 added relevant therapeutic miRNAs and key target genes.

- In addition, it was better to add forms of these miRNAs in blood of DMD such as exosome-capsulated or RNA-binding miRNAs.

Ans) Taking page limitations into account, we focused on circulating miRNAs specific to DMD and DMD cardiomyopathy or DMD female carriers. In addition, recent studies related to exosomal miR-399 and DMD cardiomyopathy were briefly mentioned in the original text and this revised manuscript.

- In addition of the miRNAs, it is better to add long non-coding RNAs corresponding to these miRNAs in DMD. 

Ans) Thank you for the valuable suggestion. However, the role of lncRNAs in DMD pathogenesis remains unclear and may not be within the scope of this review.

However, there are reports of some lncRNAs (e.g., lnc-31 and lnc-MD1) that act as myogenic regulators, and understanding the lncRNAs-miRNAs-mRNAs network would be important to explore the molecular mechanism of DMD. This revised manuscript contained a brief description/discussion about the lnc-MD1 in Part 2.3. A new reference was also added (i.e., Cesana et al. A Long Noncoding RNA Controls Muscle Differentiation by Functioning as a Competing Endogenous RNA. 2011, Cell, 147: 358-369).